# Exploration of Homocysteine Metabolism and Genetics in Autism Spectrum Disorder

**DOI:** 10.3390/nu17233761

**Published:** 2025-11-29

**Authors:** Melissa Rouphael, Tania Bitar, Hugo Alarcan, Perla Gerges, Yonna Sacre, Christian R. Andres, Walid Hleihel

**Affiliations:** 1Department of Biology, Faculty of Arts and Sciences, Holy Spirit University of Kaslik, Jounieh P.O. Box 446, Lebanon or melissa.roufael@etu.univ-tours.fr (M.R.); perla.gerges@usek.edu.lb (P.G.); walidhleihel@usek.edu.lb (W.H.); 2UMR Inserm 1253 Ibrain, Université de Tours, 37032 Tours, France; hugo.alarcan@univ-tours.fr; 3Department of Nutrition and Food Sciences, Faculty of Arts and Sciences, Holy Spirit University of Kaslik, Jounieh P.O. Box 446, Lebanon; yonnasacre@usek.edu.lb

**Keywords:** Autism Spectrum Disorder (ASD), metabolic dysregulation, *MTHFR* variants, homocysteine, vitamin B9, vitamin B12

## Abstract

Background: Understanding the genetic and metabolic profiles of individuals diagnosed with Autism Spectrum Disorder (ASD) is important for clarifying the biological characteristics of this complex disorder. Objectives: Given the limited data available for the Lebanese population, this case–control study aimed to investigate the association between common *MTHFR* variants and ASD risk and to examine differences in homocysteine metabolism between Lebanese individuals with ASD and neurotypical controls. Methods: From June 2022 to June 2023, 86 individuals with ASD and 86 controls matched for age and sex were recruited. Genotyping of the rs1801133 and rs1801131 variants and biochemical measurements were performed, followed by comparative statistical analyses. Results: Our results showed no significant association between the rs1801133 or rs1801131 variants and ASD risk (*p* > 0.05). However, the sample size was not sufficient to rule out small genetic effects. Metabolic analyses revealed significantly higher homocysteine concentrations and lower vitamin B9 levels in the ASD group (*p* < 0.0001), while vitamin B12, fasting glucose, and lipid profiles did not differ significantly between groups (*p* > 0.05). Among individuals with ASD, the TT genotype of rs1801133 was associated with elevated homocysteine concentrations (OR = 9.10, *p* = 0.014), whereas neither *MTHFR* variant was associated with vitamin B12 or B9 levels in ASD or control participants. Conclusions: Future research directions could focus on exploring the role of key enzymes associated with hyperhomocysteinemia in individuals with ASD and on replicating these preliminary findings in larger, adequately powered cohorts.

## 1. Introduction

Autism Spectrum Disorder (ASD) is a complex neurodevelopmental disorder marked by persistent difficulties in social interaction, communication, and repetitive and stereotypical behaviors or interests [1]. These challenges usually appear during the early years of childhood and last throughout life [2]. Currently, about 1 in 100 individuals globally is affected by ASD [3], with males being diagnosed more frequently than females [4]. Despite the growing awareness and recognition of ASD, diagnosing this disorder remains problematic and relies only on observational assessments as there are no validated laboratory tests available [5].

The causes of ASD are intricate and multifactorial, involving a combination of genetic, neurological, environmental, and immunological factors [6,7]. This complexity underscores the urgent need for reliable biomarkers that can help diagnose and understand ASD better. Hence, identifying such biomarkers is essential for enabling early identification and targeted interventions [8].

Recent research has focused on identifying potential biomarkers for ASD, with particular interest in homocysteine, vitamin B9, and B12 concentrations [9,10]. Homocysteine, a sulfur amino acid, plays an essential role in various biological processes, including nucleotide synthesis, cellular homeostasis, and cell cycle progression [11]. Its metabolism is dependent on the demethylation of methionine and requires cofactors such as vitamin B9 and B12 [12], which are essential for normal neurodevelopment and cognitive functions [13,14]. Individuals with ASD often exhibit selective eating habits, leading to deficiencies in these vitamins. Altun et al. (2018) and Yektaş et al. (2019) have consistently found lower concentrations of vitamin B9 and B12 in individuals with ASD compared with control groups [10,11,12,13,14,15]. These deficits can have a notable impact on individuals with ASD, resulting in delays in development and increased levels of irritability [16]. Furthermore, high concentrations of homocysteine have been associated with neurotoxic effects [17], triggering oxidative stress, and impairing mitochondrial function, both of which play a role in ASD development [18].

Additionally, different factors such as nutritional deficiencies (e.g., low vitamin B12 or folate concentrations) or genetic predispositions (e.g., *MTHFR* variants) can contribute to elevated homocysteine concentrations by impairing the metabolism of the folate–methionine cycle [19]. Methylenetetrahydrofolate reductase (MTHFR), an enzyme critical for folate metabolism, DNA synthesis, and methylation, plays a key role in converting homocysteine to methionine, as illustrated in Figure 1 [20,21]. Mutations and polymorphisms in *MTHFR* have been the focus of several studies. The *MTHFR* gene, situated on the short arm of chromosome 1 at 1p36.3 [22], presents two common variants that encode a decrease in MTHFR enzyme activity [23]. These genetic variations consist of the *MTHFR* rs1801133 variation, characterized by a change from cytosine to thymine at position 677 of exon 4. This substitution replaces alanine with valine, resulting in a thermolabile enzyme with decreased function [24]. The rs1801131 variant in the *MTHFR* gene is, located on exon 7. This variation replaces adenine with cytosine, resulting in glutamic acid being substituted with alanine at position 1298 and causing a significant reduction in enzyme activity. However, it is considered to have less impact than the rs1801133 variant [25]. While numerous studies have examined the correlation between the *MTHFR* rs1801133 and rs1801131 variants and susceptibility to ASD, the findings remain inconclusive [26,27].

The folate cycle starts with changing dietary folate, also referred to as vitamin B9, into dihydrofolate (DHF), which then converts into tetrahydrofolate (THF) through the action of dihydrofolate reductase (DHFR). Serine donates a one-carbon atom that is used by serine hydroxymethyltransferase (SHMT) to convert THF to 5,10-methylene-THF. Subsequently, methylenetetrahydrofolate reductase (MTHFR) reduces 5,10-methylene-THF to 5-methyl-THF (5-mTHF). In the methionine cycle, 5-mTHF donates a methyl group to convert homocysteine to methionine via methionine synthase (MS), which requires vitamin B12 as a cofactor. Methionine is transformed into SAM through the action of methionine adenosyltransferase (MAT), and SAM is then demethylated to produce S-adenosylhomocysteine (SAH). SAH is deadenylated and transformed back into homocysteine, thus finishing the methionine cycle. Modifications in metabolite levels linked to this process may act as potential biomarkers and targets for treating ASD [28].

Besides genetic factors, recent evidence suggests that disturbances in metabolic pathways, particularly glucose and lipid metabolism, may contribute to the pathophysiology of ASD [29,30]. Glucose serves as the brain’s primary energy source and is tightly controlled to maintain optimal neurological function. Proper glucose metabolism is essential for brain function, and disturbances in this process have been linked to various neurological disorders that affect both brain and overall health [31]. Increased glucose concentrations have been associated with adverse effects on the nervous system and cognition. These detrimental effects are caused by various mechanisms such as neuronal apoptosis, disrupted energy metabolism, synaptic dysfunction, oxidative stress in brain tissue, and impaired neuronal insulin signaling [32]. Moreover, several cholesterol abnormalities have been observed in patients with ASD [33], including a notable incidence of hypocholesterolemia [34] and decreased concentrations of triglycerides and LDL [30]. Cholesterol plays a key role in maintaining brain functions, as it is involved in processes such as synaptogenesis, myelinization, neuronal receptor functionality, and overall brain development [35,36]. Changes in cholesterol concentrations can lead to neuroinflammation, oxidative stress, and disturbances in myelination and synaptogenesis, which affect neurological functions [33]. Therefore, understanding the interaction between plasma metabolic markers and ASD is paramount for elucidating underlying mechanisms and developing targeted interventions.

Although several previous studies have investigated the *MTHFR* variants and metabolic profiles in ASD, the findings are still inconclusive [34,37,38]. Therefore, the main objective of our research was to investigate the prevalence of *MTHFR* polymorphisms, especially variants rs1801133 and rs1801131, in ASD patients compared to neurotypical controls, and to evaluate their possible association with ASD status. Furthermore, we aimed to examine plasma metabolic profiles in both ASD subjects and their matched controls. Hence, this study aimed to advance our comprehension of the etiology of ASD by combining genetic and metabolic data, paving the way for individualized interventions and targeted therapies.

## 2. Materials and Methods

### 2.1. Ethical Considerations

This research adhered to the ethical principles specified in the Declaration of Helsinki of 1964 and its subsequent revisions. Prior to implementation, the study protocol was extensively reviewed and approved by the Ethics Committee of the Holy Spirit University of Kaslik (EC 90010141). Stringent measures were implemented during the entire study to prevent participants from experiencing any physical or psychological harm. Moreover, the participants’ credentials were kept anonymous, and the study results were used only for academic purposes.

### 2.2. Participants and Study Design

A total of 172 individuals aged 3 to 18 years were recruited from all districts of Lebanon. The cohort included 86 individuals diagnosed with ASD and 86 healthy neurotypical controls. These control individuals showed no motor or language development delays or behavioral problems as reported by parents or caregivers and recorded in medical histories. They were carefully matched to the ASD group based on age, gender, and geographic location. ASD subjects were selected from non-governmental organizations (NGOs) and specialized centers, whereas controls were recruited from both private and public schools. The inclusion criteria were as follows: (1) individuals aged between 3 and 18 years, (2) those diagnosed with ASD according to the criteria of the Diagnostic and Statistical Manual of Mental Disorders, Fifth Edition (DSM-5), (3) those not taking regular vitamin supplementation, and (4) those providing informed consent from parents or caregivers. Conversely, the exclusion criteria were: (1) individuals aged below 3 or above 18 years, (2) those with Down syndrome or fragile X syndrome, (3) those taking regular vitamin supplementation, and (4) those not providing informed consent to participate in the study.

During data collection, additional information was obtained from parents or caregivers regarding potential confounding factors. Parents were asked whether their child followed any specific or restrictive therapeutic diet (e.g., gluten-free/casein-free) and whether the child was receiving any psychotropic medication. Information on gastrointestinal (GI) symptoms, such as constipation, diarrhea, abdominal pain, and nausea, was also recorded through parent report.

### 2.3. Recruitment Process

Data gathering started in June 2022 and concluded in June 2023. All headmasters of educational institutions received letters of invitation and project descriptions. After consenting, parents or caregivers were informed about the aims and methods of the study. Once they approved to participate in the research, we met with them to gather details on the child’s age, gender, and address.

#### 2.3.1. Anthropometric Measurements

All anthropometric measurements were conducted in the morning by a trained healthcare professional. Height was measured to the nearest 0.1 cm using a wall-mounted stadiometer, and weight to the nearest 0.1 kg using a calibrated digital scale, with participants barefoot and in light clothing. Body mass index (BMI; kg/m^2^) was calculated and converted to BMI-for-age z-scores (BMIZs) according to CDC growth standards for individuals aged 2–20 years. Based on BMIZ values, participants were classified as underweight, healthy weight, overweight, or obese.

#### 2.3.2. Biochemical Measurements

Peripheral venous blood was collected from all participants for analysis in the fasting state, between 7:30 and 11:00 AM. All blood samples were obtained after an overnight fast of approximately 8–12 h. Parents were contacted by phone the evening before the blood draw to remind them of the fasting instructions, and fasting compliance was verified on the morning of sampling by parental confirmation. Once the whole-blood samples were collected, they were left undisturbed to clot for a minimum of 30 min at room temperature. The sera were then separated by centrifugation 1789· *g* for 10 min at room temperature to obtain clear serum samples for analysis.

Serum lipid concentrations, including total cholesterol, triglycerides (TGs), high-density lipoprotein (HDL), and low-density lipoprotein (LDL), as well as homocysteine (Hcy), glucose, vitamin B12, and vitamin B9, were measured at two separate laboratories. Both laboratories operated under accredited quality standards, performed routine daily calibration and used manufacturer-supplied internal controls. To ensure comparability, all biochemical values were converted to a single conventional unit system. Specifically, parameters originally reported in g/L were converted to mg/dL so that all lipid and glucose measurements were expressed using the same units across laboratories. Inter-laboratory variability for each biochemical parameter was evaluated using Kruskal–Wallis tests based on the reference interval associated with each result. In accordance with these findings, all biochemical variables were analyzed using categorical classifications (low, normal, high) defined by the laboratory-specific reference intervals.

Glucose, cholesterol, and triglyceride concentrations were analyzed using enzymatic assays, while LDL and HDL concentrations were determined using a colorimetric method. Vitamin B12, vitamin B9, and homocysteine concentrations were quantified through chemiluminescent assays. The reference intervals for each parameter, along with the methods and the instruments used for their determination, are detailed in the Appendix A.

#### 2.3.3. Feeding Behavior Assessment

The Brief Autism Mealtime Behavior Inventory (BAMBI) is a validated instrument designed to assess food-related challenges and mealtime challenges in children with ASD. Originally developed by Lukens and Linscheid, BAMBI comprises 18 items grouped into three domains: limited food variety, food refusal, and features of autism. Participants respond using a 5-point Likert scale that spans from “never” to “almost every meal”, with higher scores indicating greater feeding difficulties. There are 4 items which require reverse calculation in the scale.

The tool has been extensively utilized across diverse populations and languages, showing robust psychometric properties, including high internal consistency (Cronbach’s α = 0.88) and test–retest reliability (r = 0.87) [39]. For this study, we employed the Arabic version of BAMBI, which had been previously translated and validated by Arabic-speaking researchers. To further ensure its suitability for our target population, a pilot test was conducted with 15 participants. Parents were interviewed to assess the clarity and comprehension of the questionnaire, and minor refinements were made based on their feedback. These adjustments ensured that the final Arabic version effectively captured mealtime behaviors of Lebanese children with ASD, enhancing its relevance and applicability to the study.

Given that many ASD participants consumed most of their meals in NGOs, schools, or specialized centers without parental supervision, collecting reliable dietary data was not feasible. Meal documentation was not standardized across institutions, staff were not trained to record portion sizes or actual intake, and feeding practices varied considerably between settings. These limitations would have produced inconsistent and unreliable estimates; therefore, food frequency questionnaires or detailed dietary records were not included in this study. Instead, BAMBI was used as a validated alternative to characterize feeding behaviors in the ASD group, providing contextual information to aid the interpretation of vitamin B9, vitamin B12, and homocysteine findings.

#### 2.3.4. Genotyping for rs1801133 and rs1801131 Polymorphisms

##### DNA Isolation from Peripheral Blood

3 mL of peripheral venous blood was collected from individuals in both groups, then placed into EDTA anticoagulant tubes and stored at −80 °C for future DNA extraction. DNA was obtained from whole blood using the QIAamp DNA Blood Midi Kit (Qiagen, Hilden, Germany) and Quick DNA Miniprep Plus Kit (Zymo Research, Irvine, CA, USA), following the manufacturers’ instructions. To ensure a high DNA concentration, the eluate containing the DNA was placed back onto the membrane, left at room temperature for 5 min, and then centrifuged at 4500× *g* for 2 min. Subsequently, DNA concentration and purity were assessed using the Multiskan Sky Microplate Spectrophotometer (Thermo Fisher Scientific, Waltham, MA, USA) inutile. Finally, the DNA samples were kept at −80 °C until further examination.

##### Polymerase Chain Reaction (PCR)

The genetic variations of *MTHFR* were evaluated by polymerase chain reaction (PCR) and Sanger sequencing. The PCR mixture included H_2_O MilliQ, 10X PCR Buffer II, forward and reverse primers, Accuprime Taq Polymerase, and genomic DNA. The PCR process began with an initial denaturation stage at 94 °C for 4 min, then denaturation at 94 °C for 30 s, annealing at 62 °C for 30 s, and extension at 68 °C for 40 s, repeated for 31 cycles. Subsequently, a last extension step was carried out at 68 °C for 6 min. The PCR amplicons were confirmed by electrophoresis on a 2% agarose gel and visualized using Gel Doc XR+ by BIORAD (See Appendix B). The primers used in the PCR were as follows:

For *MTHFR* rs1801133 variant (C/T):Forward primer: 5′-CCTCTCCTGACTGTCATCCC–3′Reverse primer: 5′-GCCTTCACAAAGCGGAAGAA–3′

For *MTHFR* rs1801131 variant (A/C):Forward primer: 5′-TACCTGAAGAGCAAGTCCCC–3′Reverse primer: 5′-ACAGGATGGGGAAGTCACAG–3′

##### Sequencing Analysis

The Sanger Sequencing technique was used for DNA sequence analysis, along with the Big Dye Terminator Cycle Sequencing Kit (Applied Biosystems Inc., Foster City, CA, USA). The same Forward and Reverse primers used for PCR amplification of the *MTHFR* variants were employed in the subsequent Sanger sequencing reactions to ensure accurate sequencing of the amplified regions. The sequenced samples were analyzed in the 3500XL DNA Analyzer (Applied Biosystems Inc., Foster City, CA, USA). The different genotypes are presented in the Appendix B.

### 2.4. Statistical Analysis

Statistical Packages for Social Science 22.0 was used for data analysis (SPSS Inc., Chicago, IL, USA). Participants’ sociodemographic data were summarized using descriptive statistics (mean ± standard deviation for continuous variables and frequencies/percentages for categorical categories). The Shapiro–Wilk test was used to evaluate the normality of data distribution. None of the variables were found to be normally distributed. Hence, Pearson’s chi-square test was used to compare these variables between the two groups. Hardy–Weinberg equilibrium was assessed for *MTHFR* gene polymorphisms, with Pearson’s Chi-square tests examining genotype distributions. Independent samples t-tests were used to analyze BAMBI scores. To assess whether potential confounding variables influenced biochemical markers, supplementary multivariable linear regressions were performed using continuous homocysteine, vitamin B12, and vitamin B9 levels as dependent variables and adjusting for BMI z-score, psychotropic medication use, gastrointestinal symptoms, and feeding behavior (BAMBI total score). Tests were deemed statistically significant if *p* < 0.05. To control for multiple comparisons, we defined one primary family of tests consisting of the eight biochemical parameters compared between ASD and TD groups. Bonferroni correction was applied to this family (adjusted α = 0.05/8 = 0.00625). All other analyses, including sociodemographic variables, anthropometric measures, gastrointestinal symptoms, medication use, BAMBI feeding domains, and exploratory genetic comparisons, represented distinct constructs and were therefore not combined into a single test family. These analyses were reported using exact *p*-values without Bonferroni adjustment to avoid excessive Type II error in this exploratory dataset. Regression models were exploratory and were not corrected for multiplicity. Haplotype frequencies for rs1801133 (C677T) and rs1801131 (A1298C) were estimated using the expectation–maximization (EM) algorithm. Four possible haplotypes (CA, CC, TA and TC) were reconstructed. Haplotype frequencies were computed for the whole sample and separately for ASD and control groups. Associations between haplotypes and ASD were evaluated by comparing each haplotype to all others using χ^2^ tests, and by reporting odds ratios (ORs) with 95% confidence intervals.

## 3. Results

### 3.1. Demographic Characteristics of Participants

A total of 172 individuals, comprising 86 ASD cases and 86 controls, were analyzed. Both the ASD and control groups showed similar distributions concerning age, gender, and geographic location, with a predominant male representation in both cohorts (86.0%). Sociodemographic characteristics of all subjects are presented in Table 1.

Anthropometric assessment showed no significant differences in mean weight or height between the groups. However, the ASD group had a significantly higher BMI-for-age z-score (BMIZ) than the TD group, with a greater proportion of children classified as overweight or obese and fewer in the healthy weight and underweight categories (Appendix C).

Gastrointestinal (GI) symptoms were significantly more prevalent in the ASD group. Children with ASD reported higher rates of abdominal pain (*p* = 0.043), diarrhea (*p* = 0.019), constipation (*p* = 0.009), nausea (*p* < 0.001), vomiting (*p* < 0.001), reflux (*p* = 0.001), and excessive gas (*p* < 0.001) compared to their typically developing peers. Regarding medication use, a significantly greater number of participants with ASD were prescribed risperidone (*p* = 0.002) and aripiprazole (*p* < 0.001), whereas none of the TD participants were receiving these medications. Importantly, none of the children in either group followed a specific or restrictive therapeutic diet, minimizing the potential impact of dietary interventions on the observed biochemical findings.

### 3.2. Determination of MTHFR Gene Mutations in Both Groups

#### 3.2.1. Hardy–Weinberg Equilibrium Test

The genotype frequencies of the rs1801133 and rs1801131 variants in both groups satisfied Hardy–Weinberg equilibrium, with no significant deviation between observed and expected counts (rs1801133: ASD group *p* = 0.891, control group *p* = 0.404; rs1801131: ASD group *p* = 0.879, control group *p* = 0.372; see Appendix B).

#### 3.2.2. Detection of rs1801133 and rs1801131 Variants

The distribution of *MTHFR* rs1801133 and rs1801131 variants of ASD and controls is detailed in Table 2. Our findings indicated no significant difference in the genotype distribution of the *MTHFR* rs1801133 variant between the ASD and control group (*p* = 0.196). Furthermore, there was no significant difference in allele frequencies between the two groups (*p* = 0.429). These results indicate that there is no substantial association between the *MTHFR* rs1801133 variant and ASD status within the studied population.

Similarly, there was no significant variation in the genotype distribution of the *MTHFR* rs1801131 variant between the ASD and control groups, as well as in the allele frequencies (*p* = 0.457 and *p* = 0.397, respectively). Overall, our findings suggest no significant association between the *MTHFR* rs1801131 variant and ASD within the studied population.

To further explore potential genetic associations, odds ratios (ORs) with 95% confidence intervals were calculated under per-genotype, dominant, and recessive models for the rs1801133 and rs1801131 variants (Table 3). For rs1801133, using CC as the reference genotype and considering T as the effect allele, none of the genetic models demonstrated a statistically significant association with ASD (dominant model: OR = 0.58, 95% CI 0.31–1.08, *p* = 0.085; recessive model: OR = 1.00, 95% CI 0.41–2.45, *p* = 0.999). Similarly, for rs1801131, using AA as the reference and considering C as the effect allele, no significant associations were observed across any genetic model (dominant model: OR = 0.68, 95% CI 0.37–1.25, *p* = 0.217; recessive model: OR = 0.86, 95% CI 0.30–2.50, *p* = 0.788).

To investigate the combined effect of the rs1801133 (C677T) and rs1801131 (A1298C) variants, haplotype frequencies were estimated using the expectation–maximization algorithm. Four haplotypes were observed (CA, CC, TA and TC), with TC being extremely rare (<1%) and not detected among ASD participants (Table 4). The CA haplotype was more frequent in the ASD group compared to controls (0.36 vs. 0.25), corresponding to a nominally significant association (OR = 1.68; 95% CI: 1.05–2.67; *p* = 0.039). However, the global haplotype distribution test was not statistically significant (*p* = 0.10), and this association did not remain significant after correcting for multiple comparisons. No other haplotype showed evidence of association with ASD.

### 3.3. Participants’ Eating Behavior

To characterize mealtime behavior and feeding challenges, BAMBI scores were compared between the ASD and TD groups(Table 5). Children and adolescents with ASD displayed significantly higher total BAMBI scores (59.78 ± 6.45) than TD participants (28.69 ± 2.89; *p* < 0.001), indicating more pronounced mealtime difficulties. All BAMBI domains were significantly elevated in the ASD group, including Limited Variety (29.36 ± 4.01 vs. 16.79 ± 2.47; *p* < 0.001), Food Refusal (16.08 ± 3.26 vs. 5.07 ± 0.39; *p* < 0.001), and Features of Autism (14.34 ± 3.09 vs. 6.83 ± 1.89; *p* < 0.001). These findings indicate that children with ASD exhibit significantly greater mealtime challenges, including food refusal, preference for a limited variety of foods, and autism-specific mealtime behaviors, compared to their TD peers.

### 3.4. Biochemical Profile Comparison

Our analysis revealed statistically significant differences in serum concentrations of homocysteine and vitamin B9 between both groups (Table 6). In fact, ASD individuals exhibited notably higher concentrations of Hcy compared to typically developing individuals (*p* = 0.00006). Additionally, there was a significant deficiency in vitamin B9 among ASD individuals compared to the control group (*p* = 0.0000028). Although vitamin B12 concentrations were lower in those diagnosed with ASD, this difference did not show statistical significance (*p* = 0.210).

While the ASD group showed a slightly higher prevalence of elevated fasting glucose concentrations, this difference falls just short of statistical significance (*p* = 0.0506). Similarly, there was a borderline difference in HDL concentrations between the ASD and control groups with a *p*-value of 0.0501. However, there was no significant difference in total cholesterol, low-density lipoprotein (LDL), and triglyceride concentrations between the two groups. After applying Bonferroni correction for the eight biochemical comparisons (adjusted α = 0.00625), the differences in homocysteine (*p*-adjusted = 0.00048) and vitamin B9 (*p*-adjusted = 0.0000224) between the ASD and control groups remained statistically significant.

### 3.5. Sensitivity Analysis

To examine the influence of potential confounders, supplementary multivariable linear regressions were conducted using continuous biomarker levels. After adjustment for BMI z-score, psychotropic medication use, gastrointestinal symptoms, and BAMBI score, ASD status was not significantly associated with homocysteine (*p* = 0.847), vitamin B12 (*p* = 0.978), or vitamin B9 (*p* = 0.120). Full regression models are provided in Appendix D.

### 3.6. Correlation of MTHFR Variants with Biochemical Parameters in ASD and Control Groups

Within the ASD group, our analysis showed a significant correlation between the TT genotype of the rs1801133 variant and higher odds of elevated homocysteine concentrations (OR = 9.10, 95% CI = 1.58–52.53, *p* = 0.014). Although statistically significant, the wide confidence interval indicates substantial imprecision, and this finding should therefore be interpreted cautiously and requires replication in larger samples (Table 7). However, none of the *MTHFR* genotypes for either the rs1801133 or the rs1801131 variants revealed significant associations with low vitamin B12 or B9 concentrations in this group (*p* > 0.05). Conversely, in the control group, none of the genotypes for either variant (rs1801133 or rs1801131) were significantly associated with high homocysteine concentrations, low vitamin B12 or B9 concentrations (*p* > 0.05).

Overall, these findings suggest that while the TT genotype of the rs1801133 variant is associated with higher concentrations of homocysteine in individuals diagnosed with ASD, *MTHFR* genotypes are not correlated with lower concentrations of vitamin B12 or B9 in either individuals with ASD or controls.

## 4. Discussion

Recently, there has been an increasing interest in exploring the genetic and metabolic profiles of individuals with ASD and their potential relevance to ASD pathophysiology. Therefore, the objective of this research was to examine the metabolic differences in Lebanese individuals diagnosed with ASD in comparison to neurotypical controls, and to investigate the associations between *MTHFR* variants and ASD in this population.

In our study, we recruited 86 individuals diagnosed with ASD from various regions across Lebanon. To ensure a balanced and representative sample, these participants were meticulously paired based on gender, age, and geographic area. Our findings showed a notable gender disparity, with a higher proportion of males (86%) compared to females (14%). This result aligns with previous research showing a higher occurrence of ASD in males compared to females [4,40].

### 4.1. MTHFR Polymorphisms and ASD Susceptibility

We did not observe a statistically significant association between the *MTHFR* rs1801133 and rs1801131 variants and ASD status. These findings suggest that, within this Lebanese cohort, these particular *MTHFR* variants are unlikely to exert a large effect on ASD susceptibility. However, our study was powered to detect relatively large differences in allele frequencies and therefore remains underpowered to identify the small effect sizes that are commonly observed for genetic contributions to ASD. As such, more modest associations cannot be ruled out, and our results should be interpreted as preliminary and hypothesis-generating rather than definitive. Our findings align with several previous studies that also reported no association between ASD and *MTHFR* variants [37,41,42,43,44,45,46,47,48]. However, conflicting results have been reported in the literature, with some studies finding significant correlations between *MTHFR* rs1801133 and rs1801131 variants and ASD [26,48,49].

To address variability in findings, it is essential to consider study design and statistical power. In our study, the sample size was calculated to detect a significant difference of 24% in allele T frequency of the *MTHFR* rs1801133 variant (64% in the ASD group versus 40% in the control group) with a power of 80%. Similarly, for the *MTHFR* rs1801131 variant, allele C frequencies of 61% in the ASD group and 37% in the control group informed the calculation, ensuring that 86 participants per group would provide sufficient power to detect differences.

Additionally, our analysis uncovered significant variations in the frequency of the *MTHFR* rs1801133 and rs1801131 alleles among the Lebanese ASD population in comparison to other populations. The incidence of the *MTHFR* T allele of rs1801133 variant in Lebanese individuals with ASD (34%) was notably lower compared to Saudi Arabia, Egypt, Brazil, the USA, and Canada. On the other hand, in Lebanon, the occurrence of the *MTHFR* T allele of rs1801133 variant was higher compared to the reported rates in Romania (28%) and Turkey (29%). Similarly, the prevalence of the C allele of rs1801131 variant in Lebanese individuals with ASD (31%) was comparable to that in Saudi Arabia (30%) but higher than in the USA (25%) and Canada (27%), and notably lower than the frequency documented in Egypt. These results underscore the variability in the frequency of the *MTHFR* rs1801133 and rs1801131 alleles across different populations, as presented in Table 8 and Table 9. This variability may reflect differences in genetic background, environmental exposures, or sociocultural factors that could influence the association between these polymorphisms and ASD susceptibility. However, the observed differences do not necessarily indicate causal relationships and should be interpreted cautiously. Additional research is needed to clarify the underlying factors driving these variations and their potential role in ASD etiology, particularly through studies that integrate genetic, environmental, and sociocultural dimensions.

### 4.2. Homocysteine and ASD

Our study revealed higher prevalence of hyperhomocysteinemia among Lebanese individuals diagnosed with ASD, consistent with previous research [15,51,52,53,54].

Homocysteine plays a crucial role in one-carbon metabolism, which supports neurodevelopment through DNA methylation, neurotransmitter synthesis, and cellular energy regulation. Disruptions in homocysteine balance during early development may therefore contribute to neurobiological and behavioral features relevant to ASD. Elevated homocysteine can also exert direct neurotoxic effects through oxidative stress, excitotoxicity, and impaired methylation capacity, making it a physiologically meaningful biomarker in ASD.

Notably, Ali et al. (2011) documented elevated fasting serum homocysteine concentrations in ASD children in Oman, alongside lower concentrations of vitamin B9 and B12 [51]. Similarly, Puig-Alcaraz et al. (2015) observed higher urine homocysteine concentrations in ASD children, correlating with the degree of communication deficits [54]. Conversely, Chen et al. (2024) reported that children with ASD had lower plasma homocysteine concentrations than healthy controls, highlighting the complexity and variability of metabolic profiles in ASD and emphasizing the need for further research [55].

Increased homocysteine concentrations have been proposed to influence neurodevelopment and brain function, through mechanisms such as oxidative stress, mitochondrial dysfunction, and methylation impairments, which may be relevant to ASD [56]. Homocysteine has the ability to induce the production of neurotoxic substances, such as cysteine sulphonic acid and homocysteic acid, which directly damage neurons and alter brain structural integrity [57]. Indeed, homocysteine and homocysteic acid act as NMDA receptor agonists, leading to increased calcium influx, which triggers neurotoxic effects. Consequently, this chain of cascade leads to irregular brain energy metabolism, changes in behavior, and neural or cognitive disruptions, particularly those related to spatial learning and memory [58].

Several studies have demonstrated that homocysteine influences multiple brain regions, notably the hippocampus, cortex, and basal ganglia [59,60]. These areas play a significant role in cognitive processing and behavior regulation [61,62].

Furthermore, homocysteine metabolism involves two distinct pathways: the methylation pathway and the transsulfuration pathway [17]. In the methylation pathway, homocysteine is recycled back to methionine by the catalytic action of methionine synthase (see Figure 1). This enzyme requires the presence of vitamin B12 as a cofactor and 5-methyl-THF, which serves as a methyl donor and is produced through the activity of MTHFR [63]. Conversely, the transsulfuration pathway irreversibly degrades homocysteine into cysteine via vitamin B6-dependent enzymes such as cystathionine β-synthase (CBS) and cystathionine γ-lyase (CTH) [64,65]. Various factors contribute to elevated homocysteine concentrations, including genetic defects in enzymes such as CBS, MTHFR, and MS, as well as deficiencies in vitamins B9 and B12 [66]. Addressing these imbalances through dietary interventions, such as increased intake of vitamin B9 and B12, may reduce the risks associated with hyperhomocysteinemia and its impact on neurodevelopment and cognitive function.

### 4.3. Vitamin B9 and B12 Concentrations in ASD

In line with prior research, our study revealed lower folate concentrations in individuals with ASD compared to healthy counterparts [4,67,68,69]. Ali et al. (2011) and Al-Farsi et al. (2012) both found lower serum and dietary folate concentrations in ASD children [4,67]. Guo et al. (2018) also discovered similar results, showing reduced folate concentrations in ASD children compared to typically developing controls [70]. Moreover, our study showed no significant disparity in serum vitamin B12 concentrations between ASD and control groups, which is consistent with the findings of Guo et al. (2018) [70]. On the other hand, various studies have indicated deficiencies in vitamin B12 among ASD patients [10,71,72]. Indeed, Yektaş et al. (2019) reported lower vitamin B12 concentrations, as well as higher concentrations of homocysteine in children with ASD [10]. These inconsistencies might result from variances in study populations, eating patterns, and genetic factors influencing vitamin B9 and B12 metabolism.

Vitamin B9 plays essential roles in neurodevelopment and overall brain function. It is involved in DNA synthesis and repair, methylation reactions, neurotransmitter synthesis, and regulation of gene expression [73]. Insufficient concentrations of this vitamin can result in hindered cognitive development, irritability, and weakness [74]. Deficiencies in this vitamin can frequently occur due to a range of factors, including genetic predispositions, malabsorption syndromes, and inadequate dietary consumption [75,76]. Numerous studies have shown that children with ASD are five times more likely to display selective feeding habits compared to typically developing children [77]. These feeding challenges often include food selectivity, food rejection, and disruptive eating behaviors, resulting in possible malnutrition and deficiencies in B vitamins such as vitamin B12 and folate [73,78]. For example, Barnhill et al. (2018) examined the food consumption of 86 children with ASD aged 2–8 years and 57 healthy counterparts, revealing that the majority of ASD children were deficient in several B vitamins, particularly vitamin B6 and folate [79]. Similarly, Julio Plaza-Diaz et al. (2021) assessed the nutritional intakes of Spanish preschoolers with ASD and discovered that their consumption of vitamins B6, B12, and folate was lower than recommended levels compared to control children [80].

In supplementary multivariable analyses using continuous biomarker values, ASD status was not independently associated with homocysteine, vitamin B12, or vitamin B9 after adjustment for BMI-for-age z-score, psychotropic medication use, gastrointestinal symptoms, and feeding behavior (BAMBI total score). These findings indicate that the group differences observed in the categorical biochemical classifications are not driven by these measured confounders and support the robustness of our primary results.

### 4.4. Potential Biomarkers and Nutritional Interventions

Homocysteine and vitamin B9 can serve as potential biomarkers for evaluating metabolic imbalances and guiding nutritional interventions in individuals with ASD. Few studies have explored the possible effects of folate supplementation in children diagnosed with ASD. Sun et al. (2016) investigated the impact of folic acid supplementation on oxidative stress and methylation cycles in children diagnosed with ASD. Their research found that administering folic acid improved the concentrations of folic acid, homocysteine, and glutathione redox metabolism [81]. Furthermore, Gowda et al. (2022) studied the effects of a combined supplementation regimen of betaine, pyridoxine, vitamin B9, and vitamin B12 on total homocysteine concentrations in a five-year-old child carrying a heterozygous *MTHFR* variant. The results indicated a notable decrease in serum homocysteine concentrations [82]. Furthermore, Adela Corejová et al. (2022) assessed the efficacy of methylcobalamin syrup of children and young adults diagnosed with ASD over a 200-day period. The findings revealed elevated concentrations of vitamin B12 in individuals with ASD; however, there were no notable modifications observed in homocysteine concentrations [83].

These findings emphasize the effectiveness of folate supplementation in positively reducing the concentrations of homocysteine in individuals diagnosed with ASD. Nevertheless, it is imperative to conduct more research to clarify the lasting impacts of these supplements on ASD symptoms, as well as their effect on homocysteine, vitamin B9, and vitamin B12.

### 4.5. Glucose and Lipid Metabolism in ASD

Our findings did not show any significant variations in fasting glucose concentrations or lipid profile (such as total cholesterol, HDL, LDL, and triglycerides) between ASD individuals and their matched controls. This is consistent with findings from studies by Dziobek et al. (2006) and Kim et al. (2010), who reported no notable differences in glucose and cholesterol concentrations between ASD and control groups [84,85]. However, there have been conflicting findings in other research studies. For example, Tolchard et al. (2018) found elevated blood glucose concentrations in individuals with ASD in comparison to their peers [86]. Conversely, Moses et al. (2014) observed that individuals with ASD had lower fasting blood glucose concentrations than healthy controls [87].

Regarding lipid profile, Ma et al. (2024) found high total cholesterol and triglyceride concentrations in children with ASD [88]. However, Benachenhou et al. (2019) and Al-Bazzaz et al. (2020) observed reduced HDL and LDL concentrations and increased hypocholesterolemia, with no significant difference in triglycerides between ASD and control groups [34,89]. Furthermore, Tierney et al. (2006) identified a greater occurrence of hypocholesterolemia in ASD individuals, emphasizing the intricate nature and complexity in lipid metabolism among the ASD community [30].

It has been suggested that disruptions in lipid metabolism may interfere with important neurodevelopmental processes and may be associated with ASD-related neurobiological changes [33]. Lipids play a pivotal role in maintaining cellular membrane integrity, controlling energy storage, and managing signaling processes within the central nervous system [90]. Imbalances in these processes may affect brain development and function, possibly resulting in symptoms of ASD. These mixed results highlight the complexity of metabolic differences in individuals with ASD and emphasize the importance of taking into account different factors, such as sample characteristics, methodologies, and potential subgroup differences, when analyzing the outcomes. Future studies should focus on understanding the underlying mechanisms driving metabolic alterations in individuals with ASD and investigate potential interventions to improve metabolic health in this population.

Our study cannot determine temporal direction or causality. All reported relationships between biochemical/genetic markers and ASD should therefore be interpreted as correlational and hypothesis-generating.

### 4.6. Limitations and Future Directions

This study has several limitations that should be considered in future studies. A major limitation is the inability to compare exact dosage values between the two laboratories due to differences in reference intervals. This variability may have obscured finer differences that could provide additional insights. Standardizing laboratory procedures or using a single laboratory in future studies would help address this issue and allow for more precise comparisons. Second, although feeding behavior was assessed using the validated BAMBI tool, detailed dietary intake data, such as food frequency questionnaires or food records, were not collected. Many ASD participants consumed meals in institutional settings without parental supervision, limiting the feasibility and reliability of parental dietary reporting. As a result, the study cannot directly associate measured vitamin concentrations with specific dietary patterns or nutrient intake levels. Future research should integrate comprehensive dietary assessments to better elucidate the nutritional contribution to altered vitamin and homocysteine concentrations in ASD. Moreover, socioeconomic status, which can influence both dietary quality and access to healthcare, was not characterized in detail, and some residual confounding from these unmeasured factors cannot be excluded, even though we adjusted for BMI, psychotropic medication use, gastrointestinal symptoms, and feeding behavior in supplementary analyses. Moreover, the role of difference enzymes involved in homocysteine metabolism, such as cystathionine β-synthase (CBS) and methionine synthase (MS), was not explored in this study. Studying the function of these enzymes may offer a better understanding of the processes causing hyperhomocysteinemia in ASD, aiding the development of targeted therapeutic interventions to address high concentrations of homocysteine and its associated neurotoxic effects in individuals with ASD. Moreover, although we performed an a priori power calculation, the allele tests lacked sufficient statistical power to detect the small genetic effect sizes that are typical for complex neurodevelopmental disorders such as ASD. The sample size was adequate only for relatively large differences in allele frequencies, and therefore our null findings cannot exclude more modest associations between *MTHFR* variants and ASD risk. To address this issue, future studies should be designed with substantially larger sample sizes or multicenter cohorts to ensure adequate power to detect small to moderate genetic effects. Lastly, further studies are needed to investigate how effective and safe it is to use vitamin B9 and vitamin B12 supplements to lower homocysteine concentrations in individuals with ASD. Controlled trials with varying doses and durations of supplementation can provide insights into the possible effects on both metabolic and clinical outcomes.

## 5. Conclusions

In conclusion, in this relatively small Lebanese case–control sample, we did not detect a statistically significant association between the *MTHFR* rs1801133 and rs1801131 variants and ASD risk. These findings suggest that these polymorphisms are unlikely to exert a large effect on ASD susceptibility in this population; however, given the limited statistical power of our study, smaller genetic effects cannot be excluded. However, individuals with ASD exhibited higher homocysteine and lower vitamin B9 concentrations compared to controls, indicating potential homocysteine metabolic disorders and folate deficiency. While vitamin B12 concentrations were also lower in the ASD group, the difference was not statistically significant. Notably, the TT genotype of the rs1801133 variant was associated with elevated homocysteine levels in individuals with ASD.

Taken together, our observations are consistent with a possible contribution of elevated homocysteine and folate deficiency to ASD pathophysiology, but they do not establish causality. They highlight the need for further exploration of metabolic pathways, including the roles of enzymes like CBS and MS, in larger and more comprehensively characterized cohorts. Targeting these metabolic abnormalities may provide new avenues for therapeutic interventions.

Future studies should include detailed dietary assessments to elucidate the relationship between nutrition and vitamin B9 and B12 levels in ASD, as well as controlled trials to assess the efficacy of folate and vitamin B12 supplementation in improving both metabolic and clinical outcomes. By addressing these gaps in adequately powered studies, we may advance our understanding of ASD and better evaluate the potential of targeted metabolic interventions.

## Figures and Tables

**Figure 1 nutrients-17-03761-f001:**
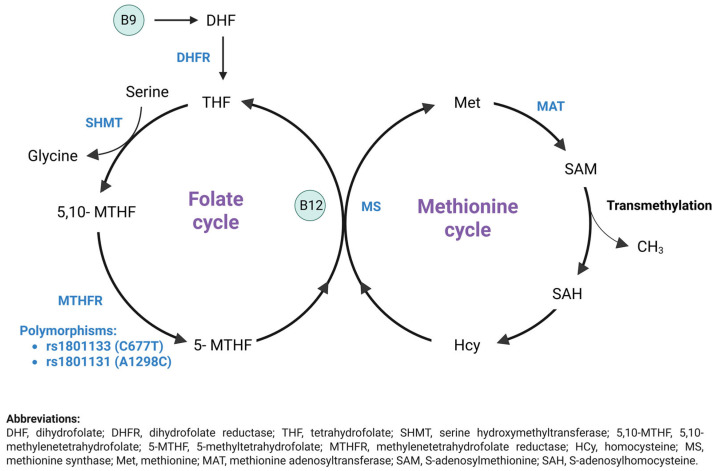
Homocysteine metabolism: Remethylation pathway (Created by Biorender) [28].

**Table 1 nutrients-17-03761-t001:** Sociodemographic characteristics of participants.

Variables	ASD Group (n = 86)	Control Group (n = 86)	*p*-Value
Age (year) ^a^ *	10.41 ± 3.51; median 10.0 (IQR 5.0); 3–18	10.41 ± 3.51; median 10.0 (IQR 5.0); 3–18	1.000
Gender ^b^ **			
Male	74 (86.0%)	74 (86.0%)	1.000
Female	12 (14.0%)	12 (14.0%)	
Geographic location ^b^ **			
Beirut	18 (20.9%)	18 (20.9%)	1.000
Bekaa	19 (22.1%)	19 (22.1%)	
North	31 (36.1%)	31 (36.1%)	
South	18 (20.9%)	18 (20.9%)	
Presence of gastrointestinal (GI) symptoms ^b^ **			
Abdominal pain	10 (11.6%)	3 (3.5%)	0.043 *
Diarrhea	22 (25.6%)	10 (11.6%)	0.019 *
Constipation	36 (41.9%)	18 (20.9%)	0.009 *
Nausea	22 (25.6%)	0 (0%)	0.000 *
Vomiting	32 (37.2%)	0 (0%)	0.000 *
Reflux	20 (23.3%)	5 (5.8%)	0.001 *
Excessive gas	14 (16.3%)	0 (0%)	0.000 *
Medication use ^b^ **			
Risperidone	9 (10.5%)	0 (0%)	0.002 *
Aripiprazole	10 (11.6%)	0 (0%)	0.0001 *

* Continuous variable is displayed as mean ± standard deviation. ** Categorical variables are described as count (n) and percentage (%). ^a^ Mann–Whitney U test. ^b^ Chi-square test.

**Table 2 nutrients-17-03761-t002:** Comparison of genotype and allelic distribution of *MTHFR* rs1801133 and rs1801131 variants between ASD individuals and healthy controls.

Genotypes\Alleles	ASD Group (n = 86)		Control Group (n = 86)		Chi-Square	*p*-Value
	Count (n)	Percentage (%)	Count (n)	Percentage (%)		
*MTHFR* rs1801133 Variant						
Genotypes						
CC	39	45.3	28	32.5	3.264	0.196
CT	36	41.9	47	54.7		
TT	11	12.8	11	12.8		
Total	86	100	86	100		
Alleles						
Allele C	57	66	52	60	0.626	0.429
Allele T	29	34	34	40		
Total	86	100	86	100		
*MTHFR* rs1801131 variant						
Genotypes						
AA	40	46.5	32	37.2	1.567	0.457
AC	39	45.3	45	52.3		
CC	7	8.2	9	10.5		
Total	86	100	86	100		
Alleles						
Allele A	59	69	55	64	0.717	0.397
Allele C	27	31	31	36		
Total	86	100	86	100		

**Table 3 nutrients-17-03761-t003:** Association analysis of rs1801133 (C677T) and rs1801131 (A1298C) under different genetic models.

SNP	Model	Comparison	OR	95% CI	*p*-Value
rs1801133 (C677T)	Per-genotype	CT vs. CC	0.55	0.29–1.05	0.071
	Per-genotype	TT vs. CC	0.72	0.27–1.89	0.503
	Dominant	(CT + TT) vs. CC	0.58	0.31–1.08	0.085
	Recessive	TT vs. (CC + CT)	1.00	0.41–2.45	0.999
rs1801131 (A1298C)	Per-genotype	AC vs. AA	0.68	0.36–1.27	0.227
	Per-genotype	CC vs. AA	0.70	0.23–2.14	0.532
	Dominant	(AC + CC) vs. AA	0.68	0.37–1.25	0.217
	Recessive	CC vs. (AA + AC)	0.86	0.30–2.50	0.788

**Table 4 nutrients-17-03761-t004:** Haplotypes analysis.

Haplotype	Overall Frequency	ASD (n = 86)	Controls (n = 86)	OR	95% CI	*p*-Value
CA	0.301	0.355	0.247	1.68	1.05–2.67	0.039
CC	0.330	0.308	0.352	0.82	0.52–1.29	0.45
TA	0.365	0.337	0.393	0.79	0.51–1.22	0.34
TC	0.004	0.000	0.009	—	—	—

Global haplotype test: χ^2^ = 4.54, *df* = 2, *p* = 0.10. Notes: ORs compare each haplotype vs. all other haplotypes combined. The TC haplotype was too rare (<1%) to allow reliable statistical testing.

**Table 5 nutrients-17-03761-t005:** Summary of BAMBI total and domain scores in ASD and TD groups.

BAMBI Score	ASD Group (n = 86)	TD Group (n = 86)	*p*-Value
Total score	59.78 ± 6.45	28.69 ± 2.89	0.000
Food refusal domain	16.08 ± 3.26	5.07 ± 0.39	0.000
Features of autism domain	14.34 ± 3.09	6.83 ± 1.89	0.000
Limited variety domain	29.36 ± 4.01	16.79 ± 2.47	0.000

**Table 6 nutrients-17-03761-t006:** Determination of biochemical parameters in both groups.

Biochemical Parameters	ASD Group (n = 86)		Control Group (n = 86)		Chi-square	*p*-Value	Adjusted *p*-Value
	Count (n)	Percentage (%)	Count (n)	Percentage (%)			
Homocysteine (μmol/L)							
Normal	61	70.9	81	94.2	16.150	0.00006 *	0.00048 *
High	25	29.1	5	5.8			
Total	86	100	86	100			
Vitamin B12 (pg/mL)							
Normal	62	72.1	69	80.2	1.569	0.210	1.000
Low	24	27.9	17	19.8			
Total	86	100	86	100			
Vitamin B9 (ng/mL)							
Normal	50	58.1	77	89.5	21.940	0.0000028 *	0.0000224 *
Low	36	41.9	9	10.5			
Total	86	100	86	100			
Fasting Glucose (mg/dL)							
Normal	78	90.7	84	97.7	3.822	0.0506	0.40480
High	8	9.3	2	2.3			
Total	86	100	86	100			
Total Cholesterol (mg/dL)							
Normal	81	94.2	83	96.5	0.524	0.469	1.000
High	5	5.8	3	3.5			
Total	86	100	86	100			
High-density lipoprotein (mg/dL)							
Normal	75	87.2	65	75.6	3.839	0.0501	0.40080
High	11	12.8	21	24.4			
Total	86	100	86	100			
Low-density lipoprotein (mg/dL)							
Normal	84	97.7	86	100.0	2.024	0.155	1.000
High	2	2.3	0	0.0			
Total	86	100	86	100			
Triglyceride (mg/dL)							
Normal	80	93.0	82	95.3	0.425	0.515	1.000
High	6	7.0	4	4.7			
Total	86	100	86	100			

Categorical variables are presented as count (n) and percentage (%). The Pearson’s Chi-square test was used to compare variables. * A *p*-value of less than 0.05 was considered statistically significant.

**Table 7 nutrients-17-03761-t007:** Logistic binary regression analysis of *MTHFR* variants and biochemical parameters in ASD and control groups.

Group	Variables	*MTHFR* Variant	Genotypes	Odds Ratio	95% Confidence Interval for OR		*p*-Value
					Lower	Upper	
ASD	High homocysteine concentrations	rs1801133	CC	1.00 Reference			
			CT	2.59	0.80	8.39	0.113
			TT	9.10	1.58	52.53	0.014 *
	High homocysteine concentrations	rs1801131	AA	1.00 Reference			
			AC	2.13	0.65	6.98	0.210
			CC	1.26	0.11	15.08	0.853
	Low vitamin B12 concentrations	rs1801133	CC	1.00 Reference			
			CT	1.63	0.56	4.77	0.373
			TT	0.26	0.03	2.68	0.260
	Low vitamin B12 concentrations	rs1801131	AA	1.00 Reference			
			AC	0.83	0.29	2.40	0.728
			CC	1.06	0.15	7.48	0.957
	Low vitamin B9 concentrations	rs1801133	CC	1.00 Reference			
			CT	1.44	0.52	3.92	0.481
			TT	2.73	0.56	13.28	0.212
	Low vitamin B9 concentrations	rs1801131	AA	1.00 Reference			
			AC	0.74	0.27	2.04	0.565
			CC	0.62	0.09	4.25	0.631
Control	High homocysteine concentrations	rs1801133	CC	1.00 Reference			
			CT	0.41	0.05	3.29	0.404
			TT	4.6 × 10^−8^	0.00	Infinit	0.999
	High homocysteine concentrations	rs1801131	AA	1.00 Reference			
			AC	7.7 × 10^−8^	0.00	Infinit	0.999
			CC	0.59	0.00	Infinit	1.000
	Low vitamin B12 concentrations	rs1801133	CC	1.00 Reference			
			CT	1.20	0.34	4.21	0.775
			TT	0.49	0.04	5.93	0.571
	Low vitamin B12 concentrations	rs1801131	AA	1.00 Reference			
			AC	1.38	0.38	5.00	0.622
			CC	0.59	0.05	7.02	0.679
	Low vitamin B9 concentrations	rs1801133	CC	1.00 Reference			
			CT	1.87	0.26	13.25	0.531
			TT	3.40	0.24	47.85	0.364
	Low vitamin B9 concentrations	rs1801131	AA	1.00 Reference			
			AC	0.94	0.16	5.61	0.946
			CC	1.76	0.10	30.09	0.697

**Table 8 nutrients-17-03761-t008:** Comparison of allele frequencies of rs1801133 variant.

Population	Year of Study	ASD Group		Control Group		*p*-Value	References
		Allele C	Allele T	Allele C	Allele T		
Romania	2009	0.72	0.28	0.75	0.25	0.60	[47]
Turkey	2014	0.71	0.29	0.76	0.24	-	[48]
Lebanon	2024	0.66	0.34	0.60	0.40	0.428	Present study
Saudi Arabia	2019	0.64	0.36	0.90	0.10	<0.0001	[26]
Egypt	2017	0.63	0.37	0.95	0.05	<0.001	[26]
Brazil	2010	0.62	0.38	0.65	0.35	0.52	[37]
Canada	2011	0.57	0.43	0.68	0.32	0.0004	[50]
USA	2004	0.49	0.51	0.68	0.32	<0.0001	[41]

(-) indicates that *p*-value was not calculated.

**Table 9 nutrients-17-03761-t009:** Comparison of allele frequencies of rs1801131 variant.

Population	Year of Study	ASD Group		Control Group		*p*-Value	References
		Allele A	Allele C	Allele A	Allele C		
USA	2004	0.75	0.25	0.68	0.32	-	[41]
Canada	2011	0.73	0.27	0.67	0.33	0.0004	[50]
Saudi Arabia	2019	0.70	0.30	0.98	0.02	<0.0001	[26]
Lebanon	2024	0.69	0.31	0.63	0.37	0.420	Present study
Egypt	2017	0.44	0.56	0.88	0.12	<0.001	[26]

(-) indicates that *p*-value was not calculated.

## Data Availability

The datasets generated and analyzed during the current study have been deposited and available in the European Variation Archive (EVA) at EMBL-EBI under accession number PRJEB96538. The direct access link is: https://eur02.safelinks.protection.outlook.com/?url=https%3A%2F%2Fwww.ebi.ac.uk%2Feva%2F%3Feva-study%3DPRJEB96538&data=05%7C02%7Cmelissa.m.roufael%40net.usek.edu.lb%7C208d14f9e5314d3ccbb808ddea28f7a9%7C06d0a2c459b94cf8a36a9190f22cd1f6%7C1%7C0%7C638924183890082463%7CUnknown%7CTWFpbGZsb3d8eyJFbXB0eU1hcGkiOnRydWUsIlYiOiIwLjAuMDAwMCIsIlAiOiJXaW4zMiIsIkFOIjoiTWFpbCIsIldUIjoyfQ%3D%3D%7C0%7C%7C%7C&sdata=XntPolAlCrnJEPIoxbzl6ST2OVw4Ku%2B9EYjCD1tXREA%3D&reserved=0 (accessed on 14 November 2025).

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
