# Peer review of "Exploration of Homocysteine Metabolism and Genetics in Autism Spectrum Disorder"

_nutrients, 2025, doi:10.3390/nu17233761_

Round 1

Reviewer 1 Report

Comments and Suggestions for Authors

The manuscript addresses a biologically plausible topic—homocysteine metabolism and MTHFR polymorphisms in ASD—in an understudied population (Lebanon). However, the manuscript has several methodological limitations, interpretative overextensions, and clarity issues that should be addressed before consideration for publication.

Major Comments

- The interpretation of vitamin deficiencies heavily relies on selective eating in ASD; however, no dietary intake data (food frequency, supplement use, feeding behaviors) were collected. Please justify this omission and acknowledge its impact on mechanistic interpretation.

- Using two labs with different reference intervals introduces potential measurement bias. Authors should describe the calibration procedures, instrumentation comparability, and how inter-lab variability was addressed statistically.

- Converting continuous values into binary categories (“normal” vs. “high/low”) reduces statistical power and may mask clinically meaningful differences. Please justify this choice and provide continuous distributions in supplementary material.

- Important variables influencing biomarkers are not collected or controlled: BMI / nutritional status; Socioeconomic status; Dietary patterns; Psychotropic medications (e.g., atypical antipsychotics); Comorbid neurodevelopmental disorders. Please discuss how these may confound associations.

- Despite power calculations, the study remains underpowered to detect small effect sizes typical of genetic contributions to ASD. Wide confidence intervals confirm instability; conclusions should be softened.

- Statements implying causality (“role in pathogenesis”) should be toned down since temporal direction cannot be established.

- Although Bonferroni correction is noted for two variables, several comparisons were run. Please specify: how many comparisons were corrected; which test families were defined; whether regression models were adjusted.

- No severity indices (e.g., CARS scores) are reported for ASD participants. Severity may correlate with biochemical alterations and should be discussed.

- Medications commonly used in ASD influence metabolism. Please clarify whether participants were medication-free.

Minor Comments

- Provide a more detailed breakdown of age distribution (median, IQR, range).

- Some sentences are redundant or overly long; light language editing is advised.

- Indicate how fasting compliance was verified (self-report vs. parental confirmation).

- Hardy–Weinberg equilibrium: consider reporting the exact p-values in the main text to improve transparency.

- Ensure all units are consistently reported (e.g., µmol/L vs. μmol/l).

- Introduce each abbreviation once; some (e.g., THF, 5-MTHF) appear before explanation in text.

- Since DSM-5 and CARS were used for inclusion, please indicate the range of CARS scores in the sample.

Author Response

We appreciate the time and effort that you have dedicated to providing your valuable feedback. We are grateful to the reviewers for their insightful comments. We have addressed all remarks point by point and revised the manuscript accordingly using track changes.

Reviewer 2 Report

Comments and Suggestions for Authors

The manuscript examines whether MTHFR variants (rs1801133, rs1801131) are associated with ASD risk, and assesses potential differences in homocysteine metabolism between Lebanese individuals with ASD and matched controls.

My comments/suggestions/questions:

  1. Please restructure the Abstract for clarity.
  2. Figure 1 would be more readable if you used full gene and marker names instead of abbreviations. It would also be helpful to annotate the specific polymorphisms studied (rs IDs) directly on the diagram.
  3. Table 2: Please add odds ratios (with 95% confidence intervals and p-values) for dominant and recessive models, as well as per-genotype odds ratios (versus the reference genotype); clearly indicate the effect and reference alleles.
  4. Please calculate haplotypes for the studied SNPs. Provide haplotype frequencies and association statistics.
  5. Please add a brief description of the physiological relevance of homocysteine in neurodevelopment and ASD, and the potential neurotoxicity of elevated homocysteine.
  6. Specify exactly which tests were adjusted for multiplicity (e.g., Bonferroni) and present adjusted p-values alongside effect sizes with 95% confidence intervals for all primary outcomes.
  7. In all tables, include measurement units and consistent reference ranges. Ensure concordant terminology.
  8. Do you have any dietary or supplementation data to help interpret B-vitamin and homocysteine differences?
  9. Do homocysteine or vitamins B9/B12 relate to ASD symptom severity or sex?

Author Response

We appreciate the time and effort that you and the reviewers have dedicated to providing your valuable feedback. We are grateful to the reviewers for their insightful comments. We have addressed all remarks point by point and revised the manuscript accordingly using track changes.
